# Evaluation of metronomic chemotherapy response using diffusion and dynamic contrast-enhanced MRI

**Mehran Baboli**[1,2]*, **Kerryanne V. Winters**[1,2], **Melanie Freed**[1,2], **Jin Zhang**[1,2], **Sungheon Gene Kim**[1,2]

**1** Center for Advanced Imaging Innovation and Research (CAI2R), Department of Radiology, New York University School of Medicine, New York, New York, United States of America, **2** Bernard and Irene Schwartz Center for Biomedical Imaging, Department of Radiology, New York University School of Medicine, New York, New York, United States of America

* mehran.baboli@nyulangone.org

**Data Availability Statement:** Data is available on Dryad: https://datadryad.org/stash/dataset/doi:10.5061/dryad.j6q573ncc.

## Abstract

### Purpose

To investigate the feasibility of using diffusion MRI (dMRI) and dynamic contrast-enhanced (DCE) MRI to evaluate the treatment response of metronomic chemotherapy (MCT) in the 4T1 mammary tumor model of locally advanced breast cancer.

### Methods

Twelve Balb/c mice with metastatic breast cancer were divided into treated and untreated (control) groups. The treated group (n = 6) received five treatments of anti-metabolite agent 5-Fluorouracil (5FU) in the span of two weeks. dMRI and DCE-MRI were acquired for both treated and control groups before and after MCT. Immunohistochemically staining and measurements were performed after the post-MRI measurements for comparison.

### Results

The control mice had significantly ($p < 0.005$) larger tumors than the MCT treated mice. The DCE-MRI analysis showed a decrease in contrast enhancement for the control group, whereas the MCT mice had a more stable enhancement between the pre-chemo and post-chemo time points. This confirms the antiangiogenic effects of 5FU treatment. Comparing amplitude of enhancement revealed a significantly ($p < 0.05$) higher enhancement in the MCT tumors than in the controls. Moreover, the MCT uptake rate was significantly ($p < 0.001$) slower than the controls. dMRI analysis showed the MCT ADC values were significantly larger than the control group at the post-scan time point.

### Conclusion

dMRI and DCE-MRI can be used as potential biomarkers for assessing the treatment response of MCT. The MRI and pathology observations suggested that in addition to the

**Funding:** This work was supported in part by grants R01CA160620, R01CA219964, UG3CA228699, and P41EB017183 from the National Institutes of Health (https://www.nih.gov/).

**Competing interests:** The authors have declared that no competing interests exist.

cytotoxic effect of cell kills, the MCT with a cytotoxic drug, 5FU, induced changes in the tumor vasculature similar to the anti-angiogenic effect.

## Introduction

Metronomic chemotherapy (MCT) is to administer conventional cytotoxic drugs in low doses frequently or continuously over extended periods without prolonged breaks [1, 2]. Previous studies have shown that MCT induces anti-angiogenic effects [3, 4] and hinders recovery of the tumor vasculature by suppressing mobilization of marrow-derived circulating endothelial progenitors (CEPs) [5]. While MCT has been shown to be a promising treatment option in various tumors, including breast cancer [6–9], prostate cancer [10], and ovarian cancer [11], determining the optimal biologic dose (OBD) of MCT for individual patients remains unresolved. Furthermore, the underlying mechanism of how a successful MCT works through balancing its anti-angiogenic and cytotoxic effects is largely unknown. The anti-angiogenic effect of MCT and its interaction with the tumor can be fundamentally different from those of VEGF-targeted therapies. We hypothesize that quantitative MRI methods could elucidate the tumor microenvironment changes induced by MCT, as MRI has been used extensively to study cancer treatment response [12–20].

Among many MRI methods, diffusion MRI (dMRI) and dynamic contrast-enhanced (DCE)-MRI are ideally suited to collecting tumor microenvironment information relevant to the underlying mechanism of MCT. dMRI measures the diffusivity of endogenous water molecules in a tissue which reflects the cellular structural properties. It is a powerful tool to detect densely populated cancer cells and their changes induced by a therapy [19, 21, 22]. In contrast, DCE-MRI has been the choice of modality to assess the perfusion properties of cancer [18]. DCE-MRI continuously measures the signal intensity change during and after contrast injection into the circulation system, which contains rich information about the tumor vasculature properties and how well blood is delivered to the tumor. These two advanced MRI methods have emerged as powerful tools for detecting early changes in vascular and cellular properties that precede morphological alterations [13, 15, 23, 24]. Since MCT regimens induce both anti-angiogenic and cytotoxic effects, it would be best to use both dMRI and DCE-MRI to assess the treatment response induced by MCT.

The purpose of the present study was to investigate the feasibility of using both dMRI and DCE-MRI to evaluate the treatment response of MCT in the 4T1 mammary tumor model of locally advanced breast cancer.

## Methods

### Animals

All mice used in this study were maintained under protocols approved by the Institutional Animal Care and Use Committee at the New York University School of Medicine. We conducted a longitudinal study on 12 female Balb/c mice (Taconic Biosciences, NY), all 6–8 weeks old at baseline. $1x10^5$ 4TI metastatic breast cancer cells suspended in 0.1 ml Dulbecco's Phosphate Buffer Saline were injected into the mammary fat pad. The mice were anesthetized with 3% isoflurane in the air for the tumor implantations and were allowed to recover and regain consciousness with the use of a heating blanket.

The mice were separated into two groups: the treatment group (n = 6) received intraperitoneal (I.P.) injections with anti-metabolite agent 5'Fluorouracil (5FU) in saline at a dose of 40mg/kg, and the control group (n = 6) was injected with an equivalent volume of normal saline on the same schedule. All I.P. injections were done on Days 7, 9, 11, 14, and 16, and MRI sessions were scheduled on Day 7 (before treatment) and 17 (24 hours after the last treatment) to compare DCE-MRI and DWI parameters before and after treatment. All mice were housed in cages with filter cage tops and, when the cage tops needed to be opened, this was performed under a hood, one cage at a time. Food and water were available *ad libitum*, and the housing room was maintained on a 12-h light-dark cycle (lights on at 07.00h). Immediately following the last MRI scan, all animals were sacrificed by trans-cardiac perfusion as described below for immunohistochemistry staining and measurements.

## MRI data acquisition

*In vivo*, MRI was performed on a 7-T Biospec micro-MRI system (Bruker Biospin MRI, Ettlingen, Germany) equipped with a volume transmitter and receiver coil. Mice were mounted on a custom 3D-printed mouse holder with temperature and respiratory monitoring probes after general anesthesia was induced using 3% isoflurane in air. During imaging, anesthesia was maintained around 1% isoflurane in air and adjusted accordingly to maintain a respiration rate of approximately 30 breaths/min. Body temperature was maintained at $34\pm2°C$ via a warm air pump system (SA Instruments, NY, USA).

Each MRI session began with a $T_2$-weighted rapid acquisition with relaxation enhancement (RARE) sequence for localization of the tumor (TR = 2 s, TE = 35 ms, voxel size = $0.125x0.125x1.5$ mm$^3$). Prior to the DCE-MRI scan, $T_1$ mapping was conducted using the RARE pulse sequence with TR = 515.4, 1085, 1884, 3236, and 10000 ms. TE = 12.3 ms, RARE factor = 4, matrix size = 320 x 180, field of view = 32 x 18 mm$^2$, 9 slices, slice thickness = 0.8 mm and inter-slice gap = 0.2 mm.

DCE imaging was performed with a $T_1$-weighted 3D FLASH sequence (TR = 12.5 ms, TE = 3.5 ms, FA = 8°, 70 repetitions, 13.6 s/frame, matrix size = 320 x 180 x 8, field of view = 32 x 18 x 8 mm$^3$, voxel size = 0.1 x 0.1 x 1.0 mm$^3$, RF spoiling) for a scan time of about 15min. A bolus of Gadolinium-diethylenetriamine penta-acetic acid (Gd-DTPA) in saline at 0.1 mmol/kg body weight was injected through a tail vein catheter, starting 3 minutes after initiating the scan.

For dMRI, a pulsed gradient spin echo (PGSE) diffusion measurement with diffusion weighting pulse width ($\delta$) = 7 ms and diffusion time ($\Delta$) = 14 ms was performed with 4-shot echo-planar imaging sequence (TR = 3 s, TE = 32 ms, field of view = 35 x 25.6 mm$^2$, Slice thickness = 1.5 mm, image matrix = 150 x 64, 8 slices, voxel size = 0.23x0.4x1.5 mm$^3$). The diffusion-weighted gradient was varied from 0–28 G/cm in the direction of [0.67, 0.67, 0.33] to have a diffusion weighting of b = 5, 39, 68, 96, 123, 177, 231, 337, 442, 546, 650, 753, and 857 s/mm$^2$ including the imaging gradients.

## MRI data analysis

The contrast-enhanced $T_1$-weighted images (last frame of DCE-MRI data) and RARE $T_2$-weighted images were used to measure tumor volume. Tumor regions of interest (ROI) were drawn manually. The DCE-MRI signal intensity *S(t)* was converted to the longitudinal relaxation rate *$R_1(t)$* using the spoiled gradient-echo sequence (SPGE) signal equation, and *$R_1(t)$* was converted to Gd concentration *C(t)* using a linear relationship with the measured baseline *$T_1$* before contrast injection and assuming the fast water exchange limit regime and longitudinal relaxivity of Gd-DTPA $r_1$ = 4.1 Lmmol$^{-1}$s$^{-1}$ [25, 26]. *C(t)* was analyzed using the modified

empirical mathematical model (EMM) [27, 28]:

$$C(t) = A(1 - e^{-\alpha t})^q e^{-\beta t} \tag{1}$$

where $A$ is the upper limit of the signal intensity, $\alpha$ (min$^{-1}$) is the rate of signal increase, $\beta$ (min$^{-1}$) is the rate of the signal decrease during washout, and $q$ is related to the slope of early uptake and the curvature of the transition from uptake to washout. Note that the exponential term with the initial washout rate $\gamma$ (min$^{-1}$) in the originally EMM is not included in Eq [1] as the EMM with $\gamma = 0$ was found adequate to describe both benign and malignant lesions in a previous study [27]. The modified EMM was used as it does not require an arterial input function, which is difficult to measure in small animal imaging studies. The maximum value of contrast concentration ($C_{max}$):

$$C_{max} \approx A(1 - \beta/(q\alpha + \beta))^q. \tag{2}$$

With the dMRI data, the apparent diffusion coefficient (ADC) was estimated by fitting a monoexponential model to the data with b-values larger than 200 s/mm$^2$.

## Immunohistochemical staining and measurements

Following the post-treatment MRI scan, mice were administered ketamine/xylazine (150/10 mg/kg, respectively) via intraperitoneal injection. Once fully sedated, the mice were transcardially perfused with phosphate-buffered saline mixed with 5,000 units/L heparin, followed by 4% paraformaldehyde (PFA). 4T1 mammary fat pad tumors were dissected following the procedure and immersed in 4% PFA overnight at 4˚C for further tissue fixation. The tumor samples were then processed for cryo-sectioning with sucrose gradients prior to embedding in optimal-compound temperature media and stored in at -80˚C until sectioned. Coronal slices were cut at 5$\mu$m for immunohistochemical (IHC) staining.

In preparation for IHC staining, heat antigen retrieval with citric acid buffer (pH = 6) was done to improve the detection of antibody staining in tumor sections. The primary antibodies used for validation of the MRI data were: Ki67 (1:250; Abcam; ab15580), CD31 (1:200; BD Bioscience; 550274) and type IV collagen (COLIV) (1:400; Abcam; ab19808). Slides were counterstained with 4', 6-diamidino-2-phenylindole (DAPI) for nuclear visualization, and cell density measurements. All slides were imaged using a whole slide digital scanner (Hamamatsu Nanozoomer 2.0HT).

The pixel size of the whole slide images was about 0.4 x 0.4 um$^2$. A patch of 500 x 500 pixels (0.2 x 0.2 mm$^2$) was used to estimate the staining density. Staining density for each antibodymarker was estimated as a number of voxels above a manually selected threshold for each antibody staining; the same threshold value was used for all images of one antibody. This approach produced staining density maps with a spatial resolution similar to those of DCE-MRI and dMRI (Fig 1).

Basement membrane was identified by COLIV staining while corresponding endothelial cells were identified by CD31. Tumor vasculature can be identified by using both CD31 and COLIV [29]. It has also shown that tumor could have immature vessels that lack COLIV immunoreactivity [30]. Fig 2 shows two example patches with different levels of COLIV expressions near the tumor vessels identified by CD31. These immature vessels could be partly due to rapid and chaotic growth of tumor vessels. In order to quantify the distribution of these immature vessels in a tumor, we calculated a covariance map for each pair of CD31 and COLIV patches. The sum of covariances near the center of the covariance map (within the distance of 3 pixels of the patch) was used as the covariance value of the patch. Pixels with CD31

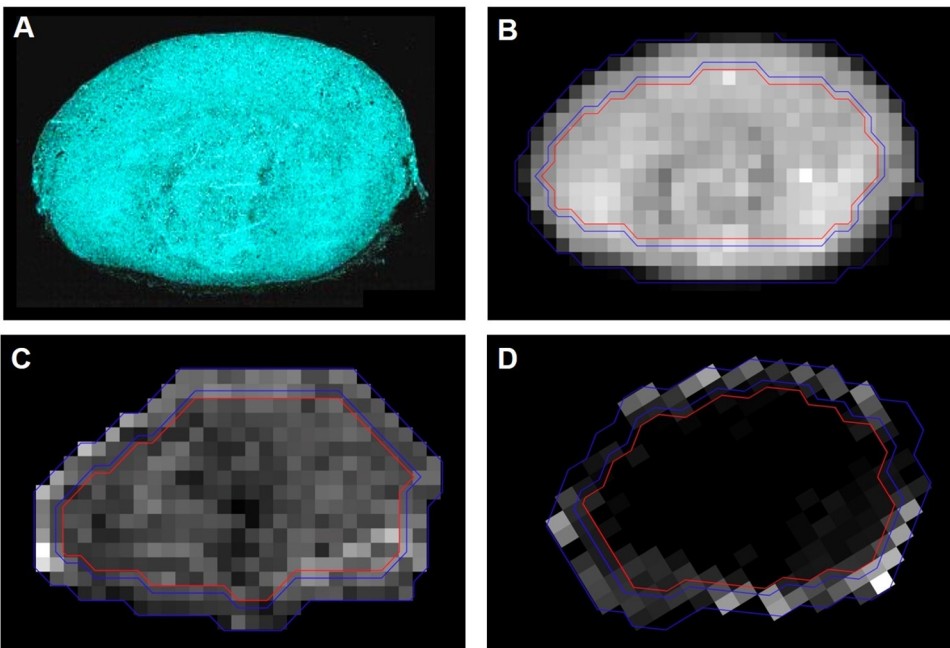

**Fig 1. Quantification of histological and MRI measures of a tumor in the control group.** (A) DAPI staining. (B) The staining density map of DAPI staining shown in (A). Also shown are the corresponding ADC (C) and $C_{max}$ (D) maps at the center of the same tumor. Two blue lines indicate a 0.2 mm-wide band immediately inside of the boundary of the tumor for the rim ROI, while the red line delineates the core ROI.

densities above the CD31 staining threshold, but below a manually selected threshold for covariance were identified as pixels with immature vessels (Fig 2C and 2D).

## Comparison with histology

The dMRI and DCE-MRI parameter maps were compared with the IHC staining density maps for the center slices, in terms of their measures in the rim and core of the tumors. This approach was taken since it is not trivial to register the IHC staining density maps to the MRI parameter maps. Regions of interest (ROI) were drawn to include the whole tumor on the dMRI, DCE-MRI, and staining density maps. The rim area of a tumor was arbitrary chosen as a 0.2 mm-wide band immediately inside of the boundary delineate by the ROI, as shown in Fig 1B. The width of the band was empirically chosen to select the thin enhancing portion of these 4T1 tumors for the rim ROI, as shown in Fig 1D.

## Statistical analysis

The difference between the control and MCT groups in terms of MRI parameters including tumor volume was assessed for each region and image parameter maps using a two-sample Student's t-test for difference of means with unequal variance. The p-values reported in this study are without a correction for multiple comparison. A p-value of less than 0.05 was considered significant. All parameter estimation and data analysis were performed using IDL (Harris Geospatial Solutions, Inc., Boulder, CO).

## Results

Tumor volume measurements were estimated at Day 7 (pre) and Day 17 (post) for all mice to determine if the MCT had an effect on 4T1 tumor growth in comparison to untreated mice

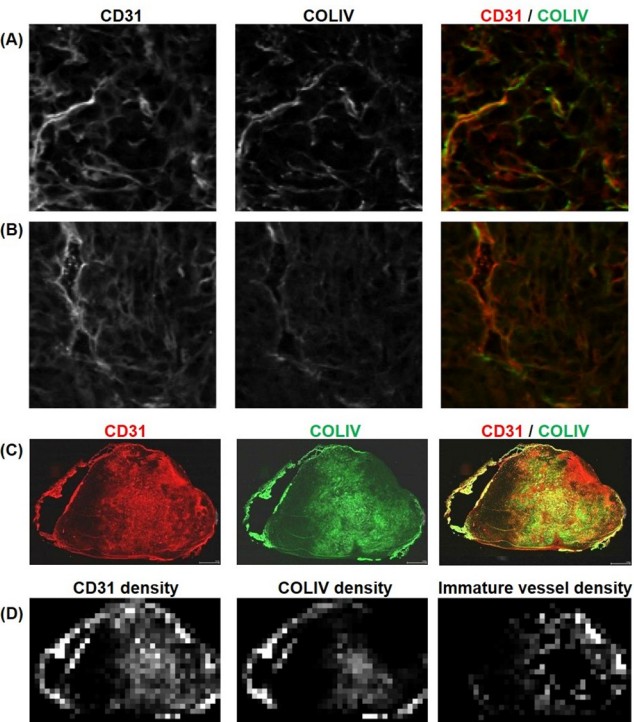

**Fig 2. Quantification of immature vessels that lack COLIV staining for basement membrane.** (A) A representative patch (500 x 500 pixels; 0.2 x 0.2 mm$^2$) with vessels stained positive mostly for both CD31 and COLIV, which has a high covariance between the two staining density maps. (B) A representative patch with vessels that has strong staining for CD31, but weak staining for COLIV. This case has a lower covariance between the two staining density maps. (C) The whole slice CD31 and COLIV stained images for a case in the control group. (D) The CD31 and COLIV density maps estimated from the corresponding IHC images shown in (C). The immature vessel density map shown in the last column was generated using the stain covariance map between CD31 and COLIV as a mask to show CD31 densities only for the patches with low covariances.

(Fig 3). Both the control (blue line) and MCT (red line) groups showed an increase in tumor volume size at the final imaging session. However, the control mice had significantly larger tumors than the MCT treated mice (p<0.005). Fig 4 shows the post-contrast DCE images and the corresponding parametric ADC mappings of the central slice of 4T1 tumors for control and MCT mouse. Following MCT, 4T1 tumors enhance in the periphery and have a higher ADC. Control tumors did not show the same enhancement pattern and had less diffusion of water.

## Treatment response in DCE-MRI

Fig 5 illustrates the differences in the average DCE enhancement in the untreated and MCT mice. Gd-DTPA concentrations and enhancement were decreased in the control group (Fig 5A), whereas the MCT mice had a more consistent enhancement between the pre-chemo and post-chemo time points (Fig 5B). Fig 6 shows the DCE empirical mathematical model parameters at pre- and post-chemo data for both groups. The amplitude of enhancement (Fig 6A) significantly differs between the untreated and MCT mice at the post scans. MCT tumors had more enhancement than the control tumors (p<0.05). A significant difference was also noted between the two groups post scans, with the MCT uptake rate being slower than the control tumors (p<0.001) (Fig 6B). Though the uptake was slower for MCT tumors, the Gd-DPA contrast was quicker to wash out than the controls, as seen by the smaller q measurements

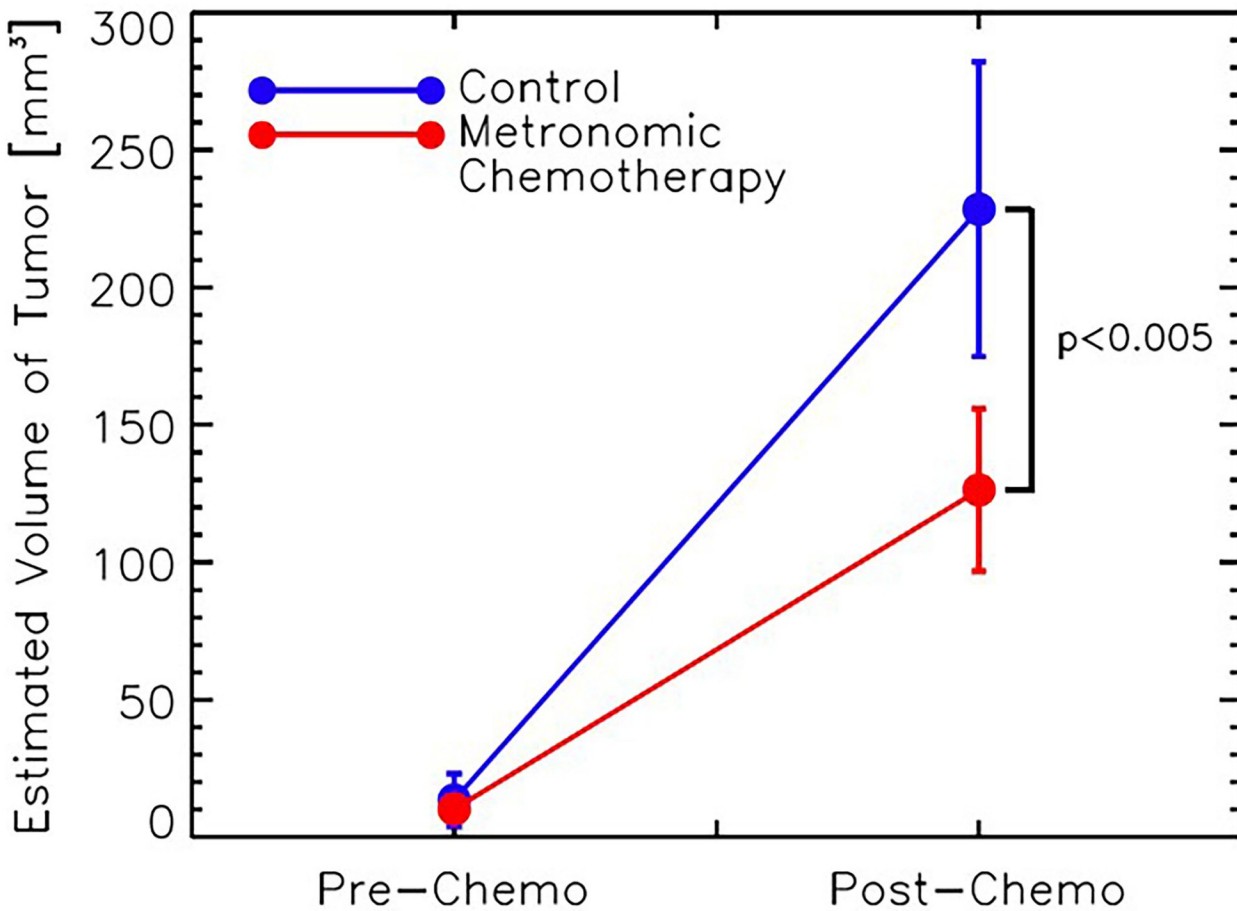

**Fig 3. Estimated tumor volume sizes compared between control (blue line) and metronomic chemotherapy-treated mice (red) at day 7 and day 17.** A significant decrease in volume size was noted in the MCT mice ($p < 0.005$).

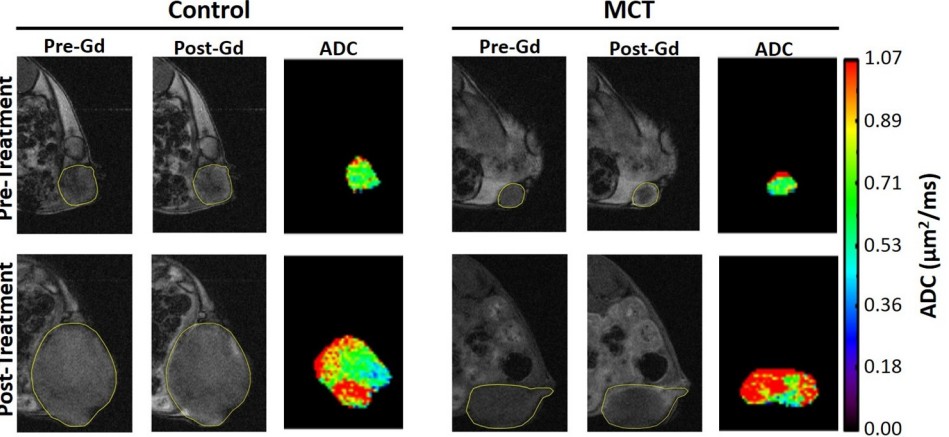

**Fig 4. Pre-contrast and post-contrast DCE images and corresponding parametric ADC mappings of the central slice of 4T1 tumors for control and MCT mouse.** An increase in ADC is seen with MCT tumors and confirms that there is a more geometrically favorable water environment with treatment. The yellow lines delineate tumor boundaries.

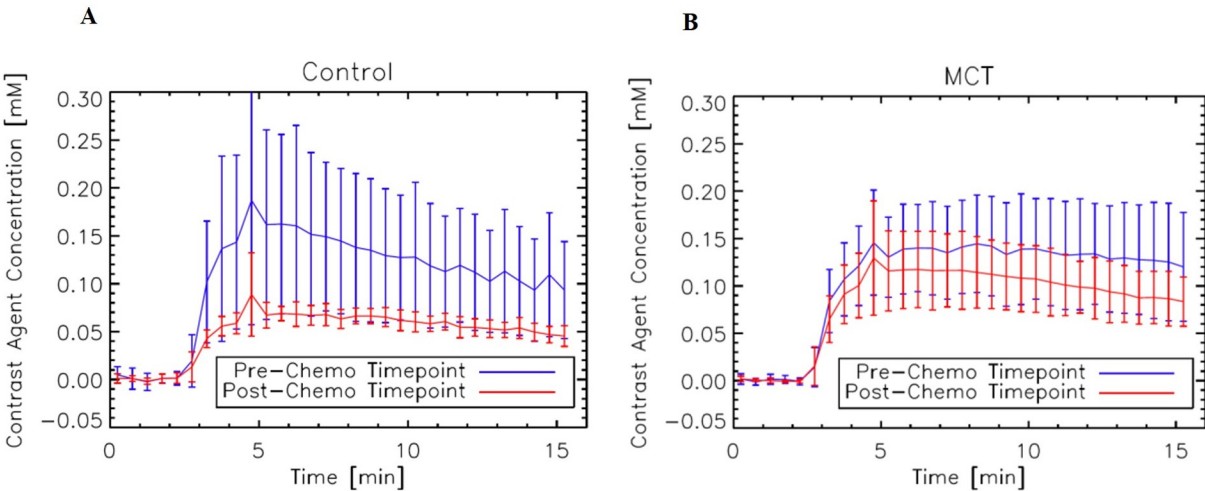

**Fig 5. Average DCE enhancement curves of control (A) and MCT (B) mice in each group pre- and post-chemo time points.** There is a decrease in enhancement with the control group, while the MCT mice remained fairly level after the scheduled chemotherapy regimen.

(p<0.005) (Fig 6C) whilere there was no significant difference in (Fig 6D). Fig 6E represents the changes in $C_{max}$ pre and post for control and MCT mice; MCT tumors had significantly higher enhancements than the controls at Day 17 (p<0.05).

## Treatment response in dMRI

Fig 7A shows the relationship between the log of the normalized MR signal and the *b*-values ranging from 0–850 s/mm$^2$. The normalized dMRI data in the log scale showed linear decay without showing a clinear indication for the effect of intravoxel incoherent motion (IVIM).

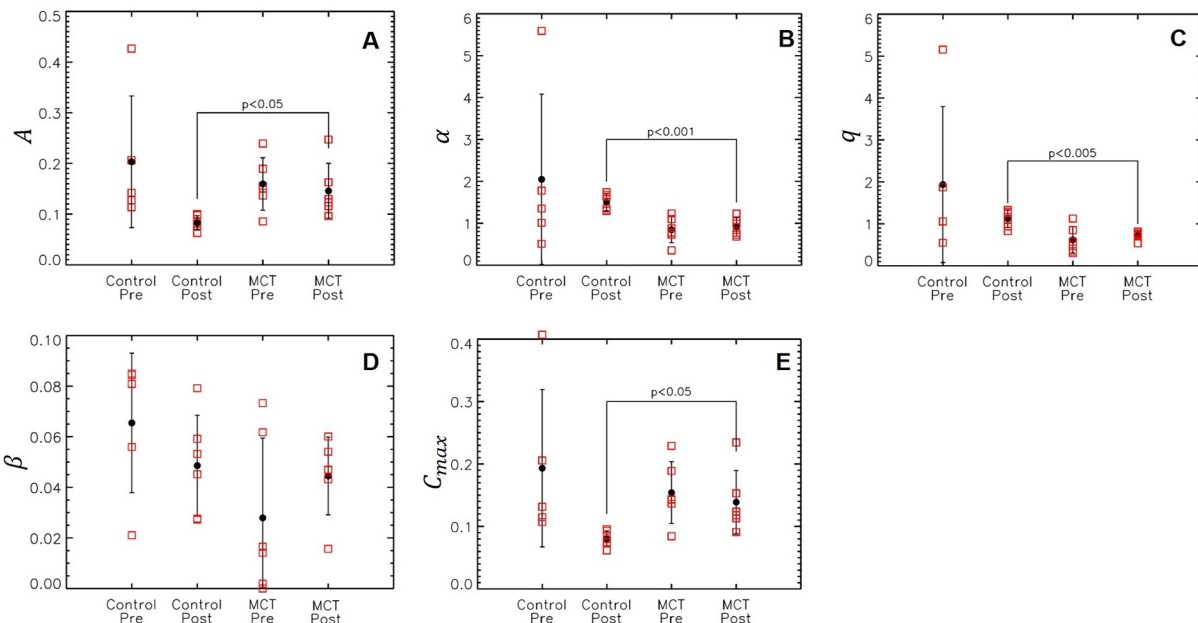

**Fig 6. DCE empirical mathematical model parameters A (A), alpha (B), q (C), β (D) and $C_{max}$ (E) in treated and untreated mice.** Significance between the two groups and *p* values shown accordingly.

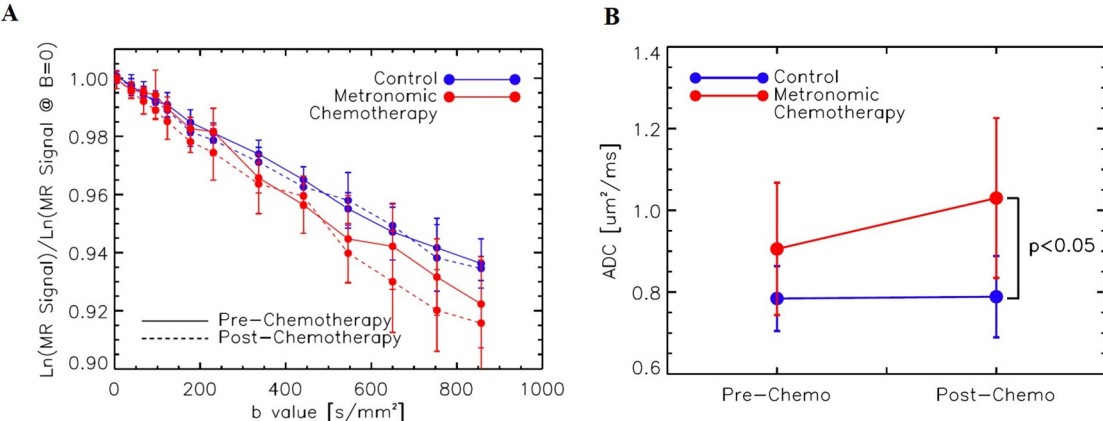

**Fig 7.** (A) Log of normalized MR signal from DWI data versus b value for MCT and control groups before and after treatment. MCT ADC values were significantly larger than the controls at the post-scan time point, which translates to the MCT tumor cellular membranes allowing for more water movement across the semi-permeable phospholipid bilayer. (B) Estimated Apparent Diffusion Coefficient (ADC) between pre and post-MRI shows an increase in diffusion with the MCT group.

Hence the analysis was done using a monoexponential model to estimate ADC. Although the IVIM effect was not clearly shown, the ADC estimation was performed by using the data with $b > 200$ s/mm$^2$, in order to minimize any potential influence of vascular properties in ADC estimation. No significant differences in ADC were noted between the control and MCT groups or between the pre- and post-scans within individual groups. An indirect relationship is present with all groups, with higher variability at larger *b*-values. As shown in Fig 7B, MCT ADC values were significantly higher than the controls at the post-scan time point.

## Histological measures

Fig 8 is a visual representation of the interpretation of immature blood vessels from the CD31 and COLIV whole slide scanning for control and MCT tumor. Immature vessel densities were computed by overlaying CD31 and COLIV whole slide images and localized vasculature networks without basement membranes.

## Comparison between MRI and histology

Fig 8 shows examples of MRI measurements were compared to histology parameters for further validation. Fig 9A shows the relationship between ADC and DAPI for the core (square) and periphery (circle) of the 4T1 tumors for the control and treated groups. In control tumors (blue), diffusion remains constant while cell number increases. MCT tumors (red) had less detectable DAPI staining and, therefore, an overall decrease in cell count following treatment and yield higher diffusion. As shown by ADC data, water molecules were more restricted in the core of the MCT tumor, while there was more movement of water molecules in the tumor rim. Both tumor groups had more proliferation at the rims; however, the MCT tumor rim was also seen to have more Ki67 staining than control tumors. Comparing the ADC and proliferation (Fig 9B) revealed that the MCT tumors had higher diffusion of water molecules than the control tumors, in addition to higher proliferative cells at the rim with higher ADC.

Both control and MCT tumor rims had a more extensive vascular network than the core, which was strongly correlated with a higher $C_{max}$ (Fig 10A). The significantly higher $C_{max}$ in MCT regimen than the controls suggests that the treatment targets the vascular endothelial cells. The rim of all tumors was composed of more immature vessels than the core for both the

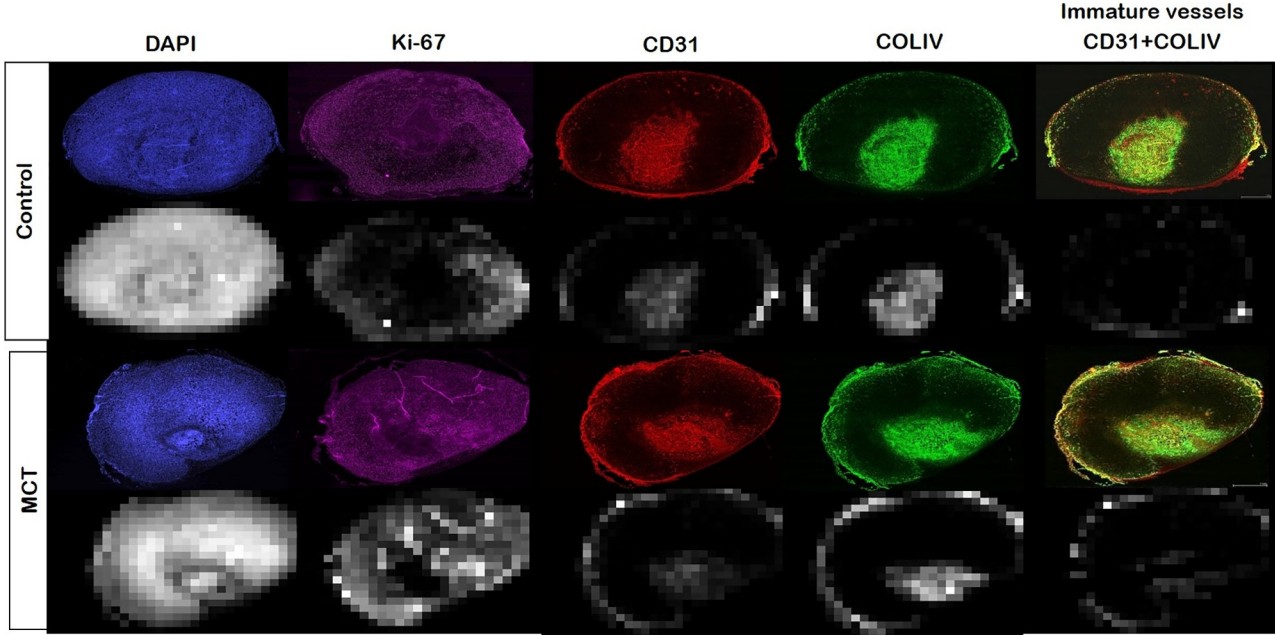

**Fig 8. Representative whole slide histology images for control and MCT mice and the corresponding staining density maps.** Immature vessels were identified by determining which vessels with CD31 expression did not have a basement membrane marker COLIV.

control and MCT mice (Fig 10B) The rims of the MCT tumors had more immature vasculature, accompanied with more local enhancement than the control group.

## Discussion

In this work, we investigated the feasibility of using dMRI and DCE-MRI to evaluate both anti-angiogenic and cytotoxic effects for the assessment of the treatment response induced by MCT. Our results showed that both DCE-MRI and DWI-MRI were able to detect changes in tumor architecture and behavior as a result of MCT treatment. The slower MCT uptake rate

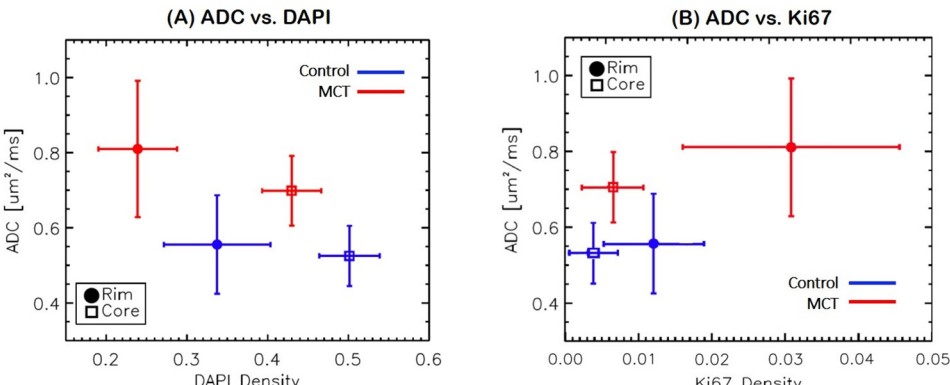

**Fig 9. Correlation between water Apparent Diffusion Coefficient (ADC) and cell density (DAPI).** Control tumors had a constant diffusion over the course of tumor progression and cell growth, while MCT mice had greater diffusion with cell death (lower cell densities in both core and rim). Ki67, a cell proliferative marker, densities are corresponding with control (blue) and MCT (red) mice in the core and periphery of 4T1 tumors. Tumor cores did not vary between the treated and untreated groups. However, the rim of the MCT mice had significantly more Ki67.

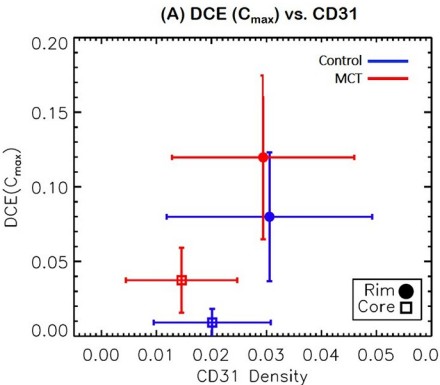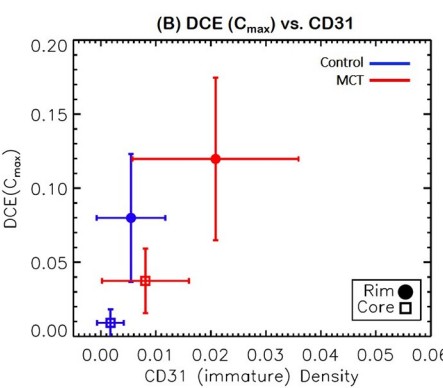

**Fig 10. Correlation between maximum CA concentration ($C_{max}$) and vasculature marker CD31 (A) and $C_{max}$ and immature vasculature (B).** Vascular density was greater in the periphery of the tumors regardless of treatment and was statistically similar between the groups; however, the MCT mice had more enhancement than the control group (A). (B) The rim of all tumors was composed of more immature vessels than the core for both the control and MCT mice. Immature vasculature was more profound in the MCT group than the control group.

observed in the DCE-MRI of the treated mice in comparison to the control tumors may be due to the less vascular bedding present with the 5FU treated mice. Moreover, the differences in the average DCE enhancement in the untreated and MCT mice confirms that it can be used as a non-invasive technique to evaluate the treatment response. The changes in the dMRI and DCE-MRI were in agreement with histology results, suggesting the feasibility of using dMRI and DCE-MRI parameters to monitor the treatment response of MCT non-invasively and to determine the OBD adaptively.

One of the interesting findings with our data is that the 4T1 tumors had almost no enhancement except the rim of the tumor. Despite such low or no enhancement in the core, the histopathology images showed no evidence of substantial necrosis in the middle of the tumor, as shown in DAPI stained image (Fig 8). More interestingly, the core regions of most 4T1 tumors typically have a distinct part with strong expression of CD31 together with parts with almost no expression of CD31. The area with strong expression of CD31 is often the central part of the core, like an island (Fig 8), surrounded by the area with the poor expression of CD31 still in the core. The part of the core with high expression of CD31 appears to match the area of slightly lower DAPI expression, suggesting this area could be becoming necrotic. It is also interesting to note that these areas with high expression of CD31 in the core also have high expression of COL-IV, suggesting they are consist of mature vessels, as opposed to newly formed vessels that could be leakier. Overall, the core of 4T1 appears to be comprised of areas either without noticeable vascularization (poor CD31 staining) or with well-developed less-leaky vessels (CD31 and COL-IV co-staining), which could explain why such low or no contrast enhancement was observed in the core regions of the 4T1 tumors in this study. It is not clear why most 4T1 tumor cores have these two types of areas with completely opposite characteristics of vascularity. Further analysis is required to understand how these areas develop and progress during tumor growth and in response to treatment.

While the histological features in CD31 and COL-IV could support the poor contrast enhancement with the 4T1 tumor, it is noted that the highly vascularized areas are not necessarily co-localized with Ki-67 staining, which is a marker of cell proliferation. As shown by the Ki-67 map in Fig 8, Ki-67 staining intensity was highest in the rim and also relatively high in the core area with poor CD31 and COL-IV staining. The ADC values in the core regions were lower than those in the rim (Fig 9A), suggesting that the non-enhancing core is not necrotic or

cystic as well. Similar patterns of contrast enhancement only in a thin band on the rim of a tumor have been observed with 4T1 tumor models in previous studies using a Gd-based contrast agent [31] as well as an iron-oxide contrast agent [32]. This pattern of distinct rim enhancement in non-necrotic solid tumors appears to be in agreement with the contrast enhancement and ADC patterns observed in inflammatory breast disease [33]. Further study is warranted to investigate how 4T1 tumors can stay as relatively solid tumors without having enough vasculature.

The diffusion MRI data in this study showed that there was no noticeable presence of the effect of intravoxel incoherent motion (IVIM) of water molecules, i.e., perfusion effect, in these 4T1 tumors in the mammary fat pad. In a previous study with 4T1 tumors in the flank, the IVIM effect was clearly observed, such that the pseudo-diffusivity and perfusion fractions were found to be correlated with the interstitial fluid pressure [34]. The present study used the same TE (35ms), diffusion pulse duration (7ms), and diffusion time (14ms) as the previous study [34]. One potential difference between the two studies is that the tumors in the mammary fat pad in the current study were on the ventral side during the imaging, whereas the flank tumors were on one side of the body. It is also not clear whether the tumor vascular characteristics could be different between tumors on the flank and the mammary fat pad. Hence, the biological differences, as well as the differences in the imaging setups, could have contributed to the discrepancy in the IVIM effect on the diffusion MRI data.

One of the interesting observations made in this tudy was that the tumors treated by MCT had similar level of contrast enhancement while the control tumors showed clearly decreaed contrst enhancement without any treatment. It is typically expected that an antiangiogenic effect is observed as reduction in contrast enhancement. However, it has also been reported that such effect could be due to "over-pruning" of the tumor vasculature [35, 36]. An adequate lower dose of antiangiogenic therapy could achieve vascular normalization where tumor perfusion can be increased or maintained for better drug delivery. As the vasculature changes still allowed for Gd-DTPA to circulate through the tumor on Day 17 better than in the tumors in the control group, this infers that the antiangiogenic effect of 5FU was at work. The tumor vasculature was functionally normalized by a weak, but still adequate amount of anti-angiogenic effect of MCT. In parallel, MCT slowed down the tumor growth and induced increased in ADC, as anticipated effect of a cytotoxic drug, compared to the control group. While this study demonstrates the feasibility of using MCT, further studies are required to investigate how to optimize MCT to maximize the cytotoxic and anti-angiogenic effects.

The DCE-MRI data in this study were analyzed by using the modified EMM of which model parameters describe the pattern of the enhancement curve without using an arterial input function (AIF). It was found nontrivial to select an arterial input function reliably from the mouse DCE-MRI data with the limited spatial resolution in this study. Since an error in the estimation of the AIF can directly propagate to the estimated contrast kinetic parameters, it was deemed that an AIF-free method, such as EMM, would be more appropriate for the present study. It is one of the limitations of the study. Future studies could be conducted with more advanced data acquisition methods based on the recent development of small animal imaging methods. The measurement of the baseline longitudinal relaxation rate constant ($T_{10}$) and transmit coil sensitivity ($B_1$) maps can be measured simultaneously with contrast kinetic parameters using the active contrast-enhanced (ACE) MRI method [37]. The $T_2^*$ effect of contrast agent can also be minimized using an ultra-short echo time pulse sequence combined with advanced imaging methods such as compressed sensing and parallel imaging [38]. These recent advances in DCE-MRI will allow estimating quantitative contrast kinetic model parameters in studies with small animal tumor models as well as patients.

Our study had other limitations. This proof-of-concept study was conducted with a relatively small cohort of animals. Non-invasive imaging techniques, such as dMRI and DCE-MRI in this study, have an important advantage of monitoring treatment response in animal models without sacrificing subsets of the animals at different time points as the feasibility is demonstrated in our study. However, we acknowledge that future studies with a larger cohort would be helpful to provide more compelling data beyond what has been shown in this proof-of-concept study. The ADC measured by a conventional dMRI method with one diffusion time can be affected by multiple factors, such as changes in extracellular space, diffusivity and cellular membrane permeability. More advanced dMRI methods with multiple diffusion times and a property model of tissue microstructure could provide more specific information about the underlying changes in the tumor [39, 40]. The cytotoxic effect of MCT was evaluated in terms of changes in DAPI and Ki67. However, the anticipated cytotoxic effect of MCT could be more directly assessed with an apoptotic marker or using high resolution H&E staining in future studies. Another limitation is that the pathology slides were obtained for only the center of the tumors and were not co-registered with the MRI data with a limited number of relatively thick slices. Due to this limitation, the data analysis was only performed in terms of the rim and core regions. More advanced approaches using a tumor-specific 3D printed mold could be used for MRI-pathology comparison within smaller regions or even at voxel levels [41, 42]. The present study was also limited to assessing the local changes in the tumor using imaging and pathological analyses and did not include any measure of related biological changes at the systemic level, such as CEPS levels. Future studies need to combine the MRI methods with pathological measures and CEPS in order to establish the imaging methods to determine the OBD of MCT.

## Conclusion

In this study, we showed the feasibility of using dMRI and DCE-MRI as potential biomarkers for assessing the cytotoxic and anti-angiogenic treatment response of the metronomic chemotherapy. The MRI and pathology observations suggested that in addition to the cytotoxic effect of cell kills, the MCT with a cytotoxic drug, 5FU, induced changes in the tumor vasculature similar to the anti-angiogenic effect. Future investigation with other types of MCT and different tumor models is also warranted to establish dMRI and DCE-MRI as a means to provide optimal treatment strategies for individual patients.

## Author Contributions

**Conceptualization:** Sungheon Gene Kim.

**Data curation:** Jin Zhang.

**Formal analysis:** Mehran Baboli, Melanie Freed, Jin Zhang.

**Funding acquisition:** Sungheon Gene Kim.

**Investigation:** Mehran Baboli, Kerryanne V. Winters, Melanie Freed, Sungheon Gene Kim.

**Methodology:** Mehran Baboli, Sungheon Gene Kim.

**Project administration:** Sungheon Gene Kim.

**Software:** Mehran Baboli, Melanie Freed.

**Writing – original draft:** Mehran Baboli.

**Writing – review & editing:** Mehran Baboli, Kerryanne V. Winters, Jin Zhang, Sungheon Gene Kim.

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
