## [Decision Letter · Decision Letter 0]

26 Jul 2020

PONE-D-20-09642

Evaluation of metronomic chemotherapy response using diffusion and dynamic contrast-enhanced MRI

PLOS ONE

Dear Dr. Baboli,

Thank you for submitting your manuscript to PLOS ONE. After careful consideration, we feel that it has merit but does not fully meet PLOS ONE’s publication criteria as it currently stands. Therefore, we invite you to submit a revised version of the manuscript that addresses the points raised during the review process.

The reviewers have raised a number of concerns regarding the methodological approach, the presentation and the statistical analysis of the study. They additionally require clarification on a number of points, including methodological rationale. The reviewers' comments can be viewed in full below.

We look forward to receiving your revised manuscript.

Kind regards,

Natasha McDonald, PhD

Associate Editor

PLOS ONE

Journal Requirements:

Additional Editor Comments (if provided):

Reviewers' comments:

Reviewer's Responses to Questions

**Comments to the Author**

1. Is the manuscript technically sound, and do the data support the conclusions?

Reviewer #1: Partly

Reviewer #2: Partly

2. Has the statistical analysis been performed appropriately and rigorously? 

Reviewer #1: No

Reviewer #2: I Don't Know

3. Have the authors made all data underlying the findings in their manuscript fully available?

Reviewer #1: Yes

Reviewer #2: Yes

4. Is the manuscript presented in an intelligible fashion and written in standard English?

Reviewer #1: Yes

Reviewer #2: Yes

5. Review Comments to the Author

Reviewer #1: Introduction

In the introduction the suppression of the mobilisation of CEPS is mentioned but not assessed, this would be an interesting addition to the manuscript to back up the vascular MRI data.

Methods

What dose of Gd-DTPA is given? Please add concentration of solution given.

The formulae for the EMM are not the same as in the referenced articles, please clarify these differences.

A separate statistics section in the methods would be advantageous. Were the data tested for normality prior to the use of t tests? Were corrections applied to the t tests to mitigate for the multiple statistical comparisons made?

Please add additional information to the figure legends so that they can be fully appreciated in isolation from the main text.

In Figure 1 what are the 3 lines on the density/MRI maps? I would have expected 2 showing the ‘rim’ region but what is the third?

Results

In the tumor volume section of the results, I think the author should be referring to tumor growth rather than cellular growth, also a reduction in tumor volume does not necessarily indicate a reduction in cell division, there could be an increase in cell death, this is not mentioned.

In figure 3 it is difficult to assess the enhancement patterns in the DCE images, it would be clearer if a pre-GdDTPA image was shown, or a representation of signal change. ROIs would also improve clarity.

The first sentence in the ‘treatment response in DCE-MRI’ section of the results is misplaced, it should be in the introduction or discussion. I think the authors are also referring to vascular normalization, so the term standardized is not correct.

An antiangiogenic effect would ordinarily be identified as a reduction in signal enhancement following Gd-DTPA administration but the authors are describing no change in contrast enhancement as ‘confiming the anti-angiogenic effect of 5FU’ but a reduction in enhancement in the control tumors as no response, please clarify these conclusions. The statement that the vascular changes ‘allow for Gd-DTPA to circulate through the tumor’ is inconsistent with the observation that enhancement is only seen in the rim of the tumors, and indeed the following sentence stating that chemotherapy cuts off the blood supply.

Please clarify how smaller q measurements relate to quicker contrast agent washout in figure 5, would showing beta values show this more clearly? The statement ‘MCT tumors had significantly more enhancements..’ is clumsy, do you refer to higher enhancement, more Gd-DTPA extravastion?

A higher ADC value cannot simply be attributed to more water being able to pass though cell membranes, cell death is often a major contributor to ADC increase. How could an increase in cell membrane permeability be assessed to back up the original explanation for ADC change?

Figure 7 does not show anything not shown in figure 8 so could be removed from the manuscript without losing any information.

Were the staining density maps for the immature vessels calculated independently from the overlaid fluorescence images or from a subtraction of COLIV from CD31? The density maps do not seem to correspond to the CD31 only staining in the fluorescence images. Please add scale bars to the histology images.

The addition of an apoptotic marker would be beneficial to demonstrate whether 5FU is having the anticipated cytotoxic effect and H&E staining would more clearly demonstrate whether there is necrosis in the tumors than DAPI alone. When assessing response to a cytotoxic agent, cell death should be properly assessed. The addition of higher power snapshots of the histology would also add to the manuscript.

In the comparison between MRI and histology section please substitute ‘with tumor growth’ with ‘in control tumors’. The lower cellular density in the MCT tumors cannot be described as a decrease unless compared to pre-treatment tumors, it should be described as being lower than in control tumors. The word ‘more’ is missing from the following sentence ‘the MCT tumor rim was also seen to have Ki76 staining than control tumors’, please also comment on the higher Ki67 staining in the core of MCT tumors.

In figure 9C please use the same scales for ADC and Ki67 as in A&B.

Discussion

Please clarify how you have come to the conclusion that the 5FU is having an anti-angiogenic effect in the rims of the MCT tumors if there are more immature vessels and higher DCE enhancement. As I understand it, the drug is likely to target the proliferating vascular endothelial cells that line immature vessels rather than the more mature vessels, so its likely that the more mature vessels would remain after treatment, and as you state in the conclusion, more mature vessels are less leaky.

You state at the beginning of the discussion that you aimed to evaluate the cytotoxic and anti-angiogenic effects of 5FU but I am not convinced that you have appropriately validated the assessment of the cytotoxic effects. No assessment of cell death have been included in the manuscript.

Please expand on why an area with high CD31 expression in the core of the tumor is likely to be becoming necrotic. You could assess the ECM to determine whether there is raised IFP in the core? This may also partially explain the lack of core enhancement if these regions are poorly perfused?

Please explain more fully how you concluded that there was no IVIM effect in these tumors, this is not explicitly mentioned in methods and analysis used monoexponential fitting.

Reviewer #2: PONE-D-20-09642

Evaluation of metronomic chemotherapy response using diffusion and dynamic contrast-enhanced MRI

Dr. Mehran Baboli

The authors have investigated the use of diffusion MRI and dynamic contrast enhanced MRI (DCE-MRI) to evaluate response to chemotherapy in a breast cancer mouse model.

The used metronomic chemotherapy induces both anti-angiogenic and cytotoxic effects. The use of both DCE-MRI and diffusion MRI is therefore well motivated.

I do have some reservations with interpretation of the data.

Specifically:

1) Is the 3D-FLASH DCE-MRI sequence RF-spoiled ? Only with RF-spoiling the sequence is purely T1 weighted enabling accurate conversion to Gd concentrations possible.

2) I think it is quite unsatisfactory that no IVIM effect was observed particularly in view of the claim of changes in the vasculature due to the anti-angiogenic treatment.

3) I don't understand the reasoning for the observed changes in the contrast enhancement in relation to (anti-) angiogenic effects. The contrast enhancement in the treated tumors decreases only slightly after treatment, to me suggesting minor changes in the tumor vasculature (at least you cannot tell from the MRI data). The chemotherapy clearly does not "cut off blood supply" as mentioned in the manuscript. For the control group a different mechanism may be at work as the tumor grows more rapidly and parts of the tumors may become less well perfusion which results in effectively lower contrast agent concentration at the later time point. In any case, the contrast enhancement does not only reflect blood supply, but also vessel permeability, which cannot be discriminated at this temporal resolution using a low-molecular contrast agent.

4) Given the observations you make. How would you discriminate a successful treatment from an unsuccessful one based on DCE-MRI and diffusion MRI, if you wouldn't have had the control group ? The DCE-MRI curves pre- and post-treatment are essentially the same (within margins of error). The same holds for the ADC values. The tumors still grow. Based on these observations alone I would conclude that the treatment does not work. So why do you claim that diffusion MRI and DCE MRI are potential biomarkers to evaluate the treatment effect ?

6. PLOS authors have the option to publish the peer review history of their article (what does this mean?). If published, this will include your full peer review and any attached files.

Reviewer #1: No

Reviewer #2: No

---

## [Author Response · Author response to Decision Letter 0]

10 Sep 2020

Please find the attached document.

---

## [Decision Letter · Decision Letter 1]

23 Oct 2020

Evaluation of metronomic chemotherapy response using diffusion and dynamic contrast-enhanced MRI

PONE-D-20-09642R1

Dear Dr. Baboli,

We’re pleased to inform you that your manuscript has been judged scientifically suitable for publication and will be formally accepted for publication once it meets all outstanding technical requirements.

Kind regards,

Domokos Máthé

Academic Editor

PLOS ONE

Additional Editor Comments (optional):

Thank you for the Authors to improve the manuscript. I agree with the second reviewer on the decision of the reader to be made assessing Figure 5 in the context of DCE and ADC to be applied as response biomarkers. The other minor comments made by Reviewer 1 are to be taken care of during the proof-reading and editing stage.

Reviewers' comments:

Reviewer's Responses to Questions

**Comments to the Author**

1. If the authors have adequately addressed your comments raised in a previous round of review and you feel that this manuscript is now acceptable for publication, you may indicate that here to bypass the “Comments to the Author” section, enter your conflict of interest statement in the “Confidential to Editor” section, and submit your "Accept" recommendation.

Reviewer #1: (No Response)

Reviewer #2: All comments have been addressed

2. Is the manuscript technically sound, and do the data support the conclusions?

Reviewer #1: Partly

Reviewer #2: Partly

3. Has the statistical analysis been performed appropriately and rigorously? 

Reviewer #1: Yes

Reviewer #2: Yes

4. Have the authors made all data underlying the findings in their manuscript fully available?

Reviewer #1: Yes

Reviewer #2: No

5. Is the manuscript presented in an intelligible fashion and written in standard English?

Reviewer #1: Yes

Reviewer #2: Yes

6. Review Comments to the Author

Reviewer #1: Thank you to the authors, the manuscript has been improved by the revisions but I still have some concerns over the conclusions made.

Despite the observation that the ADC increased and cell density decreased in the MCT tumours, in the absence of evaluation of cell death, I struggle to accept that the treatment is having a truly cytotoxic effect when proliferation is higher in the treated tumours.

It is also odd that the rim of the treated tumours show the highest level of proliferation but also the highest ADC, I don’t believe this is a demonstration of the histology and MRI biomarkers being in agreement with each other as stated in the discussion. Please discuss these observations more fully.

Stating that ‘vasculature changes still allowed for Gd-DTPA to circulate through the tumor’ is slightly inaccurate seeing as the enhancement pattern of both treated and control tumours is restricted to the outer rim of the tumour.

Please comment on figure 8 in which it would appear control tumours have fewer immature vessels than treated tumours, the removal of immature vessels is generally accepted as the mechanism of vascular normalisation.

You state that ‘dMRI and DCE-MRI can be used as potential biomarkers for assessing the treatment response of MCT’, but what would the biomarker of response to treatment be? A reduction in ADC is often observed in a responding tumour but based on these results I certainly don’t think DCE-MRI is an appropriate method, would you accept no change in DCE parameters to be a reliable marker of treatment response? There would be no control patient to compare to in a clinical setting.

In the abstract results section I think it is a little strong to say that the stable enhancements ‘confirms’ the anti-angiogenic effects, the authors state themselves that this is a proof of concept study which has a number of limitations, ‘suggests’ may be a more realistic term to use.

Please give more detail on how a covariance map was calculated for the histological assessment.

I am slightly concerned that there is some tissue that is not being evaluated between the inner rim ROI and the core ROI as shown in figure 1. Please could you clarify?

There is a lot of variation within the cohorts in figures 9 and 10, have you assessed the correlation between the MRI and histological parameters on a tumour-by-tumour basis?

Minor comments

There are a number of spelling mistakes and other errors in the text, largely in the edited parts, for example Bruker Biospin, sacrificing rather than scarifying in first line of last page of results.

In the Histological measures results section I think there is an error in the final sentence as it does not make sense should commutated be computed?

In the discussion it states that Ki67 maps are shown in figs 7 and 8 but in this version it is just fig 8.

First sentence of following section should be Fig. 9 and the edit has made the sentence not scan properly.

Suggest reconsidering the use of the term ‘stained positive mostly for both CD31 and COLIV’ in figure 2 legend.

Reviewer #2: I thank the authors for their elaborate responses to the reviewers' questions and comments.

I believe all the comments have been addressed adequately, although the question whether the anti-angiogenic drug effect of the drug can evaluated using the proposed combination of DCE-MRI and diffusion MRI remains open, given figure 5. But I guess the readers can form their own opinion.

7. PLOS authors have the option to publish the peer review history of their article (what does this mean?). If published, this will include your full peer review and any attached files.

Reviewer #1: No

Reviewer #2: No

---

## [Editor Report · Acceptance letter]

11 Nov 2020

PONE-D-20-09642R1 

Evaluation of metronomic chemotherapy response using diffusion and dynamic contrast-enhanced MRI 

Dear Dr. Baboli:

I'm pleased to inform you that your manuscript has been deemed suitable for publication in PLOS ONE. Congratulations! Your manuscript is now with our production department. 

Kind regards, 

on behalf of

Dr. Domokos Máthé 

Academic Editor

PLOS ONE